# U2 snRNA structure is influenced by SF3A and SF3B proteins but not by SF3B inhibitors

Veronica K. Urabe[1], Meredith Stevers[1], Arun K. Ghosh[2], Melissa S. Jurica[1,3]*

**1** Department of Molecular Cell and Developmental Biology, University of California, Santa Cruz, California, United States of America, **2** Department of Chemistry and Department of Medicinal Chemistry, Purdue University, West Lafayette, Indiana, United States of America, **3** Center for Molecular Biology of RNA, University of California, Santa Cruz, California, United States of America

* mjurica@ucsc.edu

**Data Availability Statement:** All relevant data are within the paper and its Supporting Information files. Raw data files for capillary electrophoresis analysis can be accessed from the following repository: 10.5281/zenodo.5527264.

## Abstract

U2 snRNP is an essential component of the spliceosome. It is responsible for branch point recognition in the spliceosome A-complex via base-pairing of U2 snRNA with an intron to form the branch helix. Small molecule inhibitors target the SF3B component of the U2 snRNP and interfere with A-complex formation during spliceosome assembly. We previously found that the first SF3B inhibited-complex is less stable than A-complex and hypothesized that SF3B inhibitors interfere with U2 snRNA secondary structure changes required to form the branch helix. Using RNA chemical modifiers, we probed U2 snRNA structure in A-complex and SF3B inhibited splicing complexes. The reactivity pattern for U2 snRNA in the SF3B inhibited-complex is indistinguishable from that of A-complex, suggesting that they have the same secondary structure conformation, including the branch helix. This observation suggests SF3B inhibited-complex instability does not stem from an alternate RNA conformation and instead points to the inhibitors interfering with protein component interactions that normally stabilize U2 snRNP's association with an intron. In addition, we probed U2 snRNA in the free U2 snRNP in the presence of SF3B inhibitor and again saw no differences. However, increased protection of nucleotides upstream of Stem I in the absence of SF3A and SF3B proteins suggests a change of secondary structure at the very 5′ end of U2 snRNA. Chemical probing of synthetic U2 snRNA in the absence of proteins results in similar protections and predicts a previously uncharacterized extension of Stem I. Because this stem must be disrupted for SF3A and SF3B proteins to stably join the snRNP, the structure has the potential to influence snRNP assembly and recycling after spliceosome disassembly.

## Introduction

The U2 small nuclear ribonucleoprotein particle (snRNP) is a vital component of the human spliceosome, which is the macromolecular complex responsible for pre-mRNA splicing during eukaryotic gene expression. U2 snRNP is composed of the U2 snRNA, core proteins, and SF3A and SF3B subcomplexes [1, 2]. During splicing, U2 snRNP is responsible for branch

**Funding:** This work was funded by National Institute of General Medical Sciences (NIGMS) of the National Institutes of Health (NIH) under awards R01GM72649 to M.S.J and A.K.G, and T32GM133391(M.S.). V.K.U. was supported by the National Science Foundation Graduate Research Fellowship under award DGE-1842400. The funders had no role in study design, data collection and analysis, decision to publish, or preparation of the manuscript.

**Competing interests:** The authors have declared that no competing interests exist.

point recognition via base-pairing interaction with the intron to select the branch point adenosine that participates in the first chemical step of splicing [3, 4]. The correct identification of the branch point sequence is important because it indirectly designates the 3′ end of the intron [5].

The structure of the first 100 nucleotides of U2 snRNA is dynamic and adopts alternate intramolecular and intermolecular conformations that have been visualized in recent cryo-EM structures models. The 17S U2 snRNP structure revealed a conformation in which the U2 snRNA adopts four consecutive stem loops: the upper region of Stem I, the Branch-interacting Stem Loop (BLS), Stem IIa, and Stem IIb [6]. Protein interactions, especially with SF3B1 and SF3B2, stabilize Stem IIa relative to the mutually exclusive Stem IIc, which forms after SF3B and SF3A proteins leave the spliceosome during catalytic activation [6]. The juxtaposition of DDX46, HTATSF1, and SF3A3 around the BSL may stabilize its structure relative to the competing lower region of Stem I [6]. Notably, the position of the BSL is incompatible with the closed conformation of SF3B1 observed in activated spliceosomes [7]. In A-complex, DDX46 and HTATSF1 are no longer present, while Stem IIa and SF3A3 remains in the same location. The BSL is replaced with the mutually exclusive extended branch helix with the intron [8]. Stem IIb rotates as the 3′ region of U2 snRNP moves to place more of SF3A in contact with the branch helix. The conformation of the first 30 nucleotides of the U2 snRNA is not modeled. In pre-B and B complex with the tri-snRNP, the 5′ end of U2 base pairs with U6 snRNA, while the branch helix, Stem IIa and Stem IIb are unchanged [9–13]. In Bact, U2 interactions with U6 increase to form the spliceosome active site [7, 14–18]. In catalytically activated spliceosomes, the branch helix docks into the spliceosome active site, while Stem IIa switches to the mutually exclusive Stem IIc that correlates with the loss of SF3B and SF3A proteins [19–34]. In all the cryo-EM structures, nucleotides that map to flexible single stranded regions between the stems and in some loops are not modeled.

Despite the bounty of information from cryo-EM structures, it is important to emphasize that the U2 snRNP structure is not rigid. There is a high degree of flexibility between the 5′ and 3′ structural modules, and the cryo-EM model reflects only one confirmation. Several open questions remain about U2 snRNA structure, chiefly how structural rearrangements are realized. For example, the topology of the branch helix necessitates an undocking of either Stem IIa or the BSL for the intron to thread between U2 snRNA and SF3A/SF3B contacts [8, 16]. When and how Stem I is unwound to enable base paring of the 5′ end with U6 snRNA is also not known. After splicing, the spliceosome disassembles, and U2 snRNA is released as part of the intron lariat spliceosome (ILS). Interactions with U6 snRNA and the intron are disrupted, and the SF3A and SF3B proteins rejoin before U2 snRNP can carry out another round of splicing. The structure of U2 snRNA during this recycling process is not known, but at some point, Stem I, the BSL and Stem IIa must reform. Finally, it is not known when or if the lower portion of Stem I that is mutually exclusive with the BSL forms and unwinds.

SF3B interacts extensively with U2 snRNA. In the 17S snRNP, the loop of Stem I is located next to the RRM1 of SF3B4, and both the BSL and Stem IIa interact with SF3B2 and SF3B1 [6]. We reasoned that SF3B inhibitors, which affect SFB1 conformation and interfere with U2 snRNP's incorporation into the spliceosome, could also affect U2 snRNA structure. The inhibitors arrest spliceosome assembly at an SF3B inhibitor-stalled complex (SI-complex) that is unstable relative to A-complex [35–37]. Previous studies suggested that the SI-complex has a branch helix, but whether it is mediated by the same nucleotides as an A-complex branch helix is unclear [38].

In this study, we used chemical probing to compare U2 snRNA structure in A-complex to SI-complex. We also examined inhibitor effects on U2 snRNA in U2 snRNP in the presence and absence of SF3A and SF3B proteins. Surprisingly, we detect no differences in the U2

snRNA secondary structure in the SI-complex compared to the functional A-complex, and the data are consistent with the presence of the top of Stem I, the branch helix, Stem IIa, and Stem IIb. From these findings, we conclude that SF3B inhibitors do not interfere with the crucial steps of BSL unwinding and branch helix formation, and that Stem I disruption is not required for those events. SF3B inhibitors also have no discernable impact on the U2 snRNA structure in U2 snRNP. However, loss of SF3A/B components results in unexpected protections in the 5′ end of U2 snRNA. Both endogenous and synthetic protein-free U2 snRNA produce a similar reactivity pattern, which structure prediction algorithms propose is due to extension of Stem I by three additional base pairs. With synthetic U2 snRNA, the extended Stem I appears quite stable and is not disrupted by single nucleotide changes. We postulate that U2 snRNA takes on this conformation upon release of U2 snRNP from the intron lariat complex after splicing. The structure may play a role in recruitment of SF3B and SF3A components back to the snRNP for another round of spliceosome assembly.

## Results

### U2 snRNA secondary structure is maintained in SF3B inhibited splicing complexes

Structural differences between recent cryo-EM models of *H. sapiens* 17S U2 snRNP [6] and *S. cerevisiae* A-complex spliceosomes [8] predict several RNA structural rearrangements. First, the BSL must unwind to allow the branch point recognition sequence of U2 snRNA to fully base pair with the intron's branch point sequence. Additionally, the 3′ half of the BSL forms ~10 base pair-like interactions with intron nucleotides upstream of the branch point. SF3B inhibitors stall A-complex between an ATP-dependent event and before full stabilization of the complex. We hypothesized that U2 snRNA will exist in an intermediate structure in the SF3B inhibited complex (SI complex).

To test this idea, we assembled spliceosome complexes in HeLa nuclear extract on a pre-mRNA substrate with a branch point sequence that is perfectly complementary to the U2 snRNA branch point recognition sequence. The pre-mRNA substrate also contains the hairpin recognition sequence for MS2 (MS2 bacteriophage coat protein), which binds tightly to the fusion protein MS2:MBP (maltose binding protein) for affinity purification. To accumulate A-complex, we depleted the nuclear extract of functional tri-snRNP using endogenous RNase H and oligonucleotides complementary to the U4 and U6 snRNA [36]. To accumulate SI-complexes, we treated nuclear extract with 1μM spliceostatin A (SSA) prior to spliceosome assembly [36]. We isolated the spliceosome complexes by size exclusion followed by amylose affinity selection for MS2:MBP [39]. The complexes were then immediately probed with either dimethyl sulfate (DMS) to modify the Watson-Crick faces of adenosine and cytosine bases or 1-methyl-7-nitroisatoic anhydride (1m7) to acylate the ribose 2′-OH. Chemically reactive nucleotides correlate with unpaired bases for DMS and RNA backbone flexibility for 1m7. Nucleotides are protected from chemical modification by base pairing or protein interactions. We mapped modifications by reverse transcriptase primer extension with an oligonucleotide complementary to nucleotides 97–117 of U2 snRNA, and extension products were analyzed by sequencing gels. Reactive nucleotides were identified as bands with consistently increased intensity relative to control reactions lacking the chemical modifiers across replicate probing reactions. The control reactions are necessary to account for reverse transcriptase stops that result from endogenous U2 snRNA modifications or by highly stable secondary structure.

The overall patterns of reactivity in A-complex and SI-complex are very similar (Fig 1). Uridines in the loop of Stem I are reactive to 1m7 while the upper part of the stem is generally protected (Fig 1A, lanes 9 & 10 vs. lanes 8 & 11). Although C13 in the stem shows some

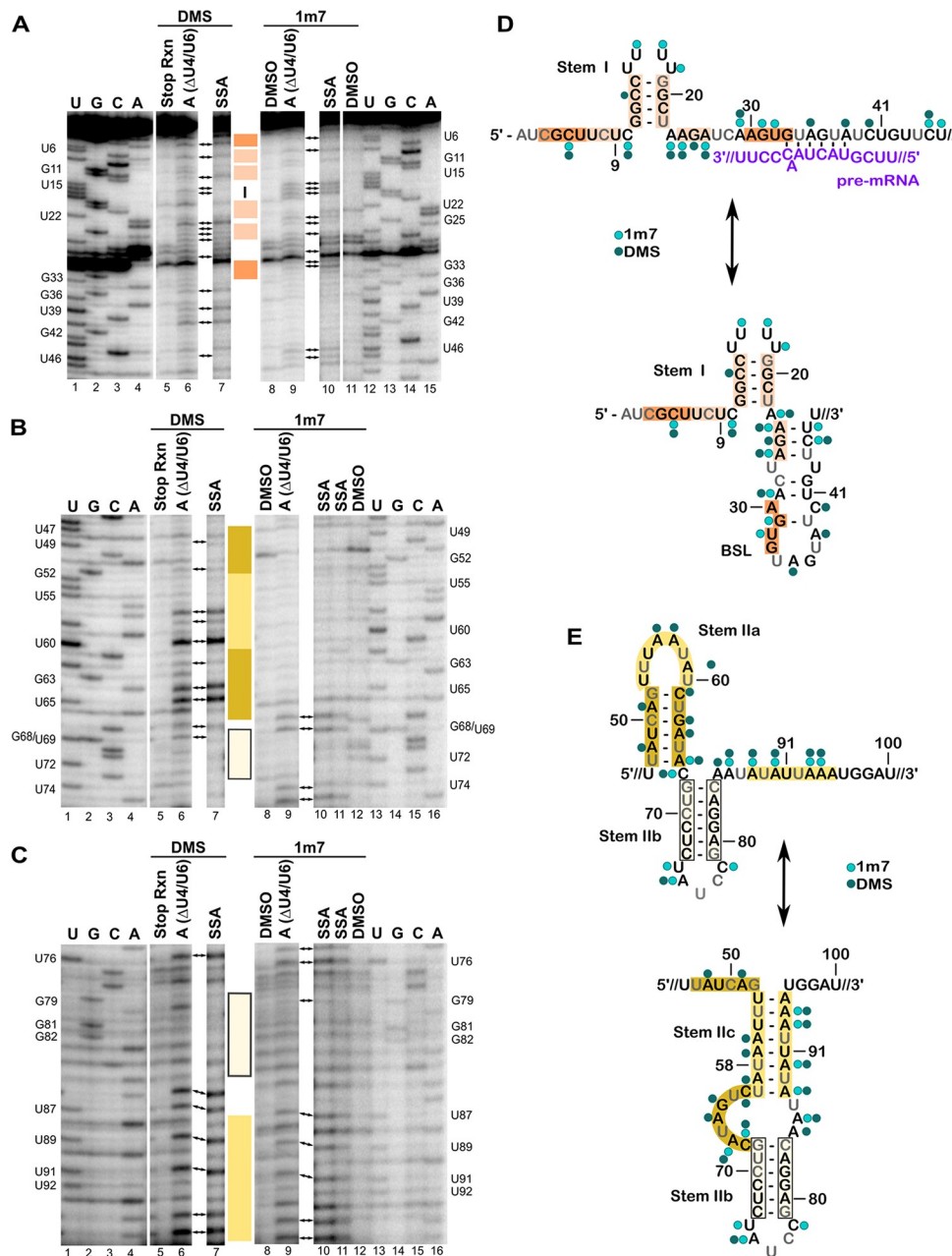

**Fig 1. Chemical probing of U2 snRNA in A-complex and SI-complex spliceosomes.** (A) Sequencing gel analysis of reverse transcription primer extension products correlating to nucleotides 1–46 of U2 snRNA isolated from purified A-complex (ΔU4/U6) and SI-complex (SSA) modified with DMS or 1m7. The radiolabeled primer is complementary to U2 snRNA nucleotides 97–117. Lanes 5 show a stop reaction with quench added before DMS, and lanes 8 and 11 represent control reactions with DMSO instead of DMS or 1m7. Parallel sequencing reactions (lanes 1–4 and 12–15) map the location of the indicated nucleotides labeled on the left and right and structural elements of U2 snRNA are denoted by colored boxes and correlate to the structure models shown in D and E. Arrows point to primer extension products observed consistently across triplicate experiments. (B & C) Same as in (A) except that nucleotides 47–74 are shown in B, nucleotides 75–95 are shown in C, and the last DMSO and sequencing lanes are shifted to 12–16. (D & E) Reactivity patterns mapped on two competing structural models of U2 snRNA (black text), with the structure most consistent with the data shown on top. Nucleotides in grey correlate with primer extension stops in controls. The circles denote reactivity to 1m7 (light green) and DMS (dark green). Intron is indicated by the purple text.

modification with DMS (Fig 1A, lanes 6 & 7 vs. 5), the overall modification pattern indicates that the top of Stem I is not unwound in either complex. Nucleotides 24–26, which can participate in either 3′ half of Stem I's lower region or the 5′ half of the mutually exclusive BSL, are reactive to both reagents, indicating the absence of both structures. DMS and 1m7 reactivity of nucleotides A29 and C45 further support the absence of the BSL. Because these reactivity patterns are present in both SI-complex and A-complex, we conclude that SSA does not interfere with BSL unwinding.

Nucleotides from the branch point recognition sequence through U44 are protected from 1m7 modification in both complexes, which is consistent with the extended branch helix duplex that appears to be stabilized by interactions with SF3A2 and SF3A3 in the cryo-EM structure of A-complex [8]. However, nucleotides A35, A38, and C40 in the same region show some limited reactivity to DMS, which is slightly enhanced in the SI-complex. This may be expected for C40 which would be opposite another cytosine residue in the pre-mRNA substrate used to assemble the complexes (Fig 1D). However, A35 and A38, are expected to base pair with the two uracil residues in the consensus branch point sequence. Combined with the 1m7 results, this result suggests that U2 snRNA structure is maintained while some introns base pairing interactions are briefly destabilized. This idea would fit the proposed fidelity check on branch helix fidelity by Prp5 [40]. Additionally, SF3B1 inhibitors are predicted to interfere with SF3B1 closing that appears to stabilize the branch helix in the A-complex cryo-EM model [41, 42]. In the context of an open SF3B1 conformation, base pairs with intron could be more unstable and again briefly expose the base pairing faces of A35 and A38 to DMS modification. Nevertheless, the general protection of nucleotides in this region supports the conclusion that SSA does not block branch helix formation.

Similarity between A-complex and SI-complexes reactivity patterns continues through the Stem IIa/IIb/IIc regions (Fig 1B and 1C). C45 and U46, which are situated in the stretch of nucleotides that links the extended branch helix and Stem IIa, are reactive to 1m7. Low reactivity of the next seventeen nucleotides (47–63) to 1m7 is consistent with the presence of Stem IIa as modeled in the S. cerevisiae A-complex (Fig 1B, lanes 9–11 vs. lanes 8 & 12). The tetraloop structure of Stem IIa that docks onto SF3B proteins explains a lack of 1m7 reactivity in this region. DMS data for nucleotides U47-A66 also support the predicted Stem IIa structure, with nucleotides A48, A51, and C61 in the stem showing weak reactivity and the unpaired A56, A57, and A59 in the U-turn loop showing stronger reactivity (Fig 1B, lanes 6 & 7 vs. lane 5). Surprisingly, both G63 and A64 at the base of the stem are highly reactive to DMS. A64 is modeled as base paired with U49, and DMS modification of guanosine at N7 is typically inefficient and does not result in a stop in the primer extension reaction. We suspect that the unusual DMS reactivity reflects some sort of structural strain may impact the local chemical environment.

Residues A66 and C67, which link Stem IIa and IIb, show the expected reactivity to both DMS and 1m7. Residues in Stem IIb G68-C73 are protected (Fig 1B, lanes 6 & 7 vs. 5, lanes 9–11 vs. lanes 8 & 12) along with their base pairing partners G79-C83 in both complexes (Fig 1C, lanes 6 & 7 vs lane 5, lanes 9–11 vs lanes 8 & 12). In contrast, U74 and A75 in the loop of Stem IIb, are highly reactive to 1m7. The remaining nucleotides in the loop yield strong stops in both experimental and control reactions and cannot be evaluated. Finally, nucleotides leading into and continuing to the 3′ half of Stem IIc (U86-A94) are generally reactive to both reagents, consistent with its mutually exclusive conformation relative to Stem IIa.

Together our data indicate that the conformation of U2 snRNA observed in the S. cerevisiae cryo-EM A-complex structure is maintained in human A-complex. We also find that branch helix formation is not blocked by the SF3B inhibitor SSA. This result is consistent with previous psoralen crosslinking studies showing U2 snRNA interaction with an intron in the

presence of inhibitor, albeit differently positioned relative to the branch point [38]. The similar reactivity patterns of A-complex spliceosomes and SI-complexes also mean that SF3B, the target of the inhibitor, does not regulate U2 snRNA structure and likely controls another aspect of spliceosome assembly to interfere with stable recruitment of the tri-snRNP for the next stage of spliceosome assembly. SI-complexes readily disassociate when challenged with the polyanion heparin as compared to A-complex [36]. In that context, branch helix formation alone may not be sufficient to stabilize U2 snRNP's engagement with the intron. Instead, SF3B1 closure over the branch helix, which the inhibitors are proposed to block, is likely required. SF3B1 closure may also signal correct branch point adenosine selection for continued spliceosome assembly. Finally, because the upper region of Stem I is intact in A-complexes arrested by either depletion of functional tri-snRNP or by the presence of SSA, we speculate that unwinding of Stem I to enable interactions with U6 snRNA when tri-snRNP is recruited likely follows SF3B1 closure.

## U2 snRNA structure in the snRNP is dependent on SF3A/B association

SF3B inhibitors proposed to stabilize SF3B1 in an open conformation, so we also tested whether SSA affects U2 snRNA structure in the snRNP by DMS probing. For our analysis, we used HeLa nuclear extract with either 150 mM potassium chloride (KCl) for 17S U2 snRNP or 420 KCl for 12S U2 snRNP, which destabilizes SF3B and SF3A protein complexes [2], both in the presence or absence of SSA. We also examined the structure of endogenous U2 snRNA in the absence of associated proteins. Nucleotide reactivity was mapped as described for A complex and the SI-complex.

Overall, the reactivity pattern for the 17S snRNP is consistent with original DMS probing studies by Behrens, et al. and the recent cryo-EM model [6, 43] (Fig 2A and 2B). Similar to our results with spliceosome complexes, we see no distinct differences in DMS reactivity with SSA treatment (S1 Fig). For the first 23 nucleotides, only C10, C14, and C21 in the top region of Stem I are in the intact U2 snRNP (Fig 2A, lane 4 v lane 3). C13 in the stem shows some reactivity, similar to what we observed in A-complex spliceosomes where the lower region of Stem I is unwound to accommodate the formation of the BSL [6]. A24, which is in the linker between Stem I and the BSL, is protected, which may reflect an interaction with SF3A3. SF3A3 is notable in that it nestles a short alpha-helix between Stem IIa and the base of the BSL, where it positions a tryptophan to stack on the last base pair of the BSL [6]. SF3A3 maintains that same position in A-complex where it stacks the same tryptophan at the end of the branch helix [8]. Limited reactivity for nucleotides A26, C28, A38, C40, and C45 in the BSL stem, along with a reactive A35 in the BSL loop indicates its presence as well. Nucleotides in Stem IIa follow the same reactivity pattern as in A-complex, including the unusual reactivity of G63. Stem IIb nucleotides are protected, except for A75 in its loop (Fig 2B, lane 2 vs lane 1). Finally, nucleotides in the 3′ half of Stem IIc are reactive as expected with the presence of the mutually exclusive Stem IIa.

For the most part, U2 snRNA in the snRNP incubated in high salt conditions exhibits a similar pattern of reactivity as the intact snRNP with two notable exceptions (Fig 2A and 2B). First, nucleotides in Stem IIa show increased reactivity that indicate its destabilization, which was also noted in previous work [43] (Fig 2B, lane 3 vs lane 2). Unexpectedly, there is no corresponding decrease in nucleotide reactivity that supports a switch to Stem IIc. In contrast, U2 snRNA probed in the absence of all proteins shows strong protections that are consistent with the switch to Stem IIc and correlate with increased reactivity of nucleotides in the mutually exclusive Stem IIa (Fig 2B, lane 4 vs lane 3). This result suggests that SF3B and SF3A components are needed to stabilize Stem IIa and that either the core snRNP or a factor in nuclear

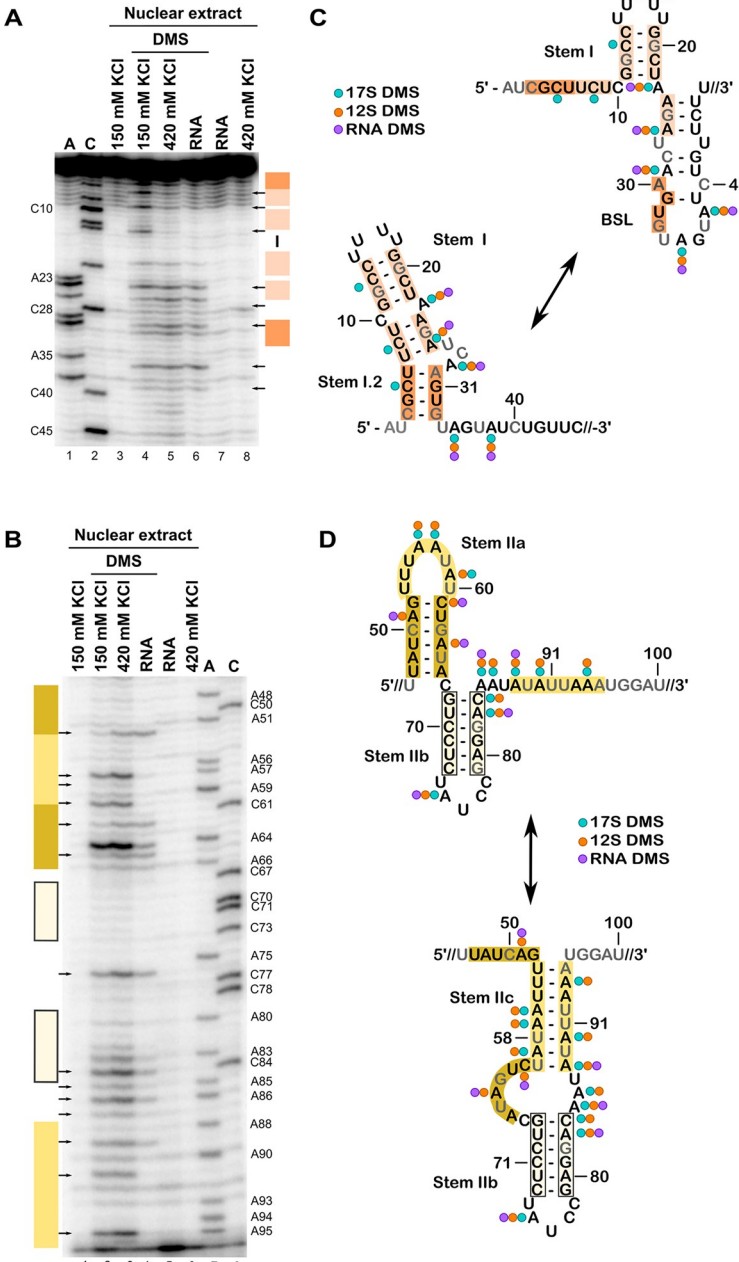

**Fig 2. Chemical probing of U2 snRNA in U2 snRNP and RNA.** (A-B) Sequencing gel analysis of reverse transcription primer extension products from U2 snRNA isolated from nuclear extract with 17S U2 snRNP (150 mM KCl) or 12S snRNP (420 mM KCl) modified with DMS. Protein-free RNA purified from nuclear extract (RNA) was also probed. The gels are labeled as in Fig 1, with DMSO control reactions in lanes 3, 7 & 8, and parallel sequencing reactions in lanes 1 & 2. Arrows point to primer extension products observed consistently across triplicate experiments. Nucleotides 1–46 are shown in A, and 47–95 in B. (C & D) Reactivity patterns mapped on two competing structural models of U2 snRNA with the structure most consistent with 17S U2 snRNP data shown on top. Nucleotides in grey correlate with primer extension stops in controls. The circles denote reactivity to DMS for 17S (green) or 12S (orange) U2 snRNPs or protein-free U2 snRNA (purple).

extract prevents Stem IIc formation. Such a factor could serve to promote the rejoining of SF3B and SF3A after U2 snRNP is released from the spliceosome.

The second difference between U2 snRNP in the lower and higher salt conditions is in the 5′ end. Nucleotides C5, C8, C10, and C14 become protected when SF3A and SF3B proteins are destabilized with higher salt (Fig 2A, lane 3 vs lane 2). C8, C10 and C14 protections were also noted in the early study by Behrens et al. [43]. Additionally, protein-free U2 snRNA exhibits a similar increase in protection (Fig 2A, lane 4 vs lane 2). The pattern is consistent with an intact Stem I. However, the expectation is that the 3′ half of the BSL would be exposed if nucleotides in its 5′ half participate in Stem I, and we do not observe a correlating increase in reactivity for the mutually exclusive BSL in A38, C40, and C45 (Fig 2C). Importantly, the same reactivity pattern is also observed in protein-free U2 snRNA, ruling out the possibility that a protein is responsible for the unexpected protections (Fig 2A, lane 4 vs lane 2). Furthermore, it suggests that the interactions responsible for the protections are intrinsic to the RNA and might not involve canonical base pairing.

## Chemical mapping of synthetic U2 snRNA predicts an extended Stem I

To further investigate the unexpected patterns of protection that we observe in U2 snRNA in the absence of SF3B and SF3A proteins, we probed a synthetic U2 snRNA construct consisting of the first 100 nucleotides of U2 snRNA flanked by two stable stem loops to use as probing standards and a 3′ tail for primer extensions (Fig 3). Notably, the synthetic RNA lacks the endogenous modifications found in U2 snRNA isolated from nuclear extract. We also probed synthetic U2 snRNA with series of mutations in the Stem I region to examine the importance of individual base pairs on the structure [44]. For the probing experiments, we treated the synthetic U2 snRNAs with 1m7, and mapped nucleotide reactivity by primer extension followed by capillary electrophoresis. Reactivities were quantified using HiTrace [45] and used as restraints for structure prediction with the RNAstructure Fold algorithm [46].

Like the reference structures, nucleotides in the loops of Stem I, Stem IIb and the 3′ half of Stem IIa (nt 61–67) of the WT synthetic U2 snRNA are highly reactive to 1m7 (Fig 4A). Nucleotides in the loop (U34-A35) of the BSL are also reactive, and while the BSL stem exhibits lower reactivity, it is not to the same degree as the highly protected nucleotides in Stem IIb, Stem IIc, and the reference stems. Notably, nucleotides at the 5′ end of the U2 sequence (C6-C10) and the 5′ half of Stem IIa are also protected. Secondary structures predicted based on the 1m7 reactivity uniformly contained Stem I, Stem IIb, Stem IIc, and the two reference stems (Fig 4A) [47]. Nucleotides in the 5′ half of Stem IIa were predicted to base pair with a synthetic buffer sequence situated between the end of the U2 snRNA and the last reference stem. The software also predicted four continuous base pairs between C3-U6 and A30-G33, which extends Stem I. This proposed extension of Stem I is mutually exclusive with the BSL,

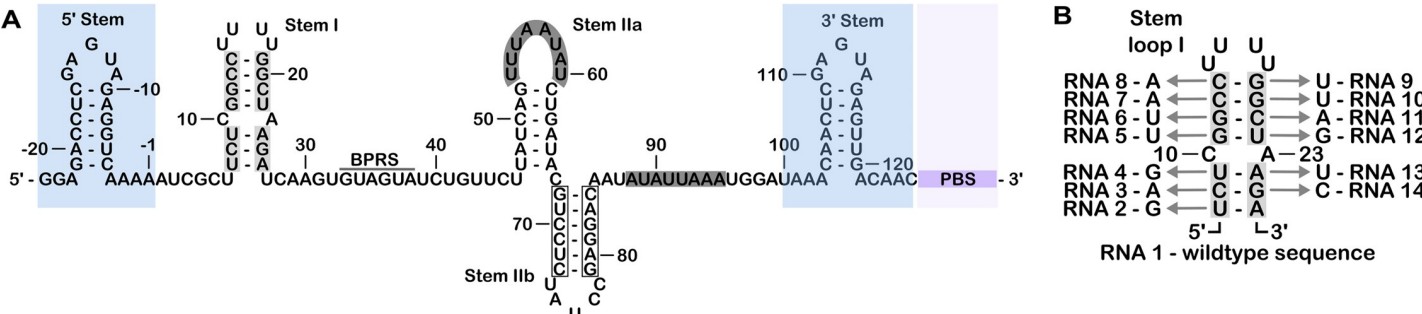

**Fig 3. Synthetic U2 snRNA and mutations to disrupt Stem I.** (A) The 5′ end of U2 snRNA flanked by the highlighted reference stems (5′ Stem and 3′ Stem) with primer binding sequence (PBS) (B) Stem I mutations and correlating construct numbers.

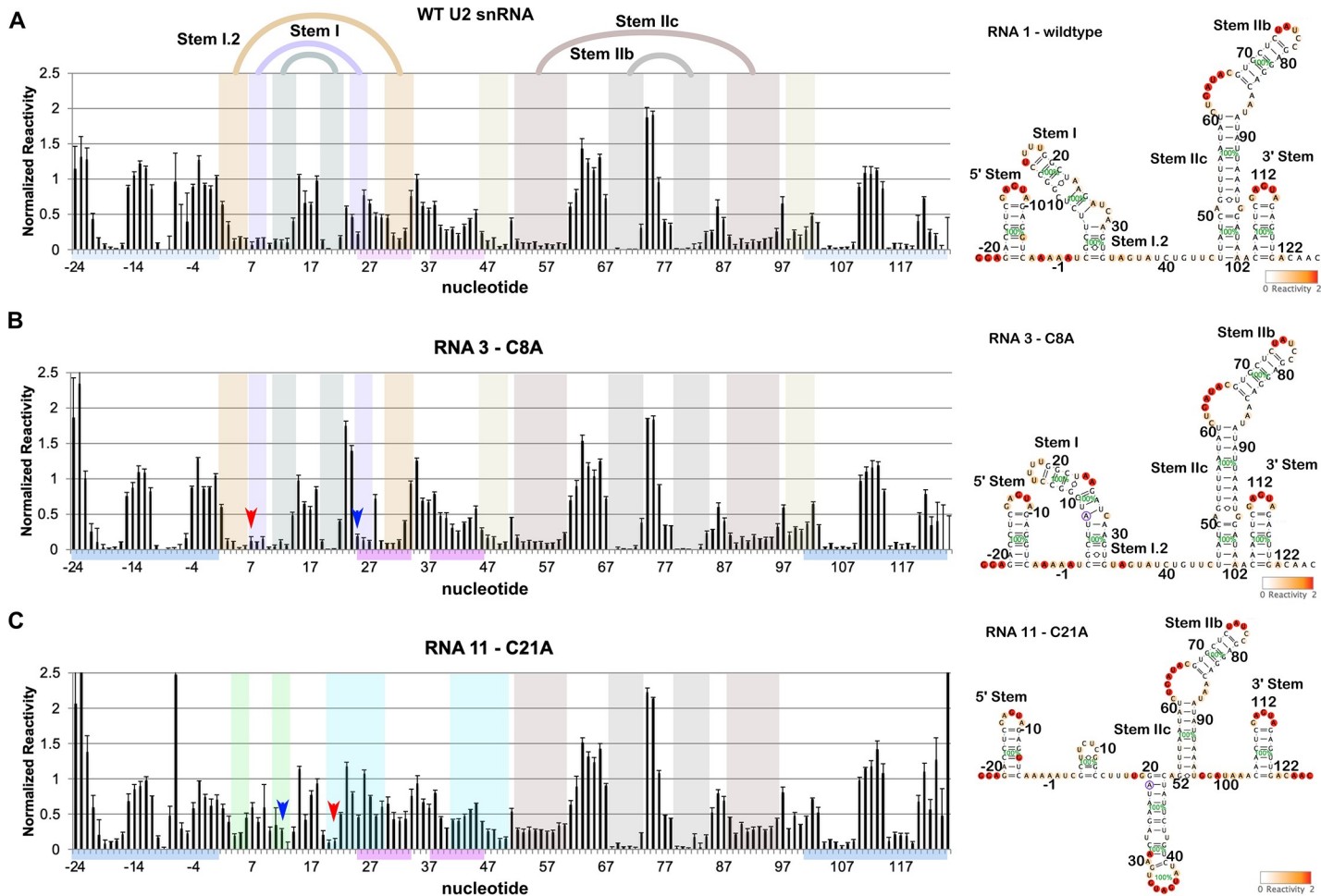

**Fig 4. Chemical probing of synthetic U2 snRNA predicts an extended Stem I.** The left panels show normalized reactivities to 1m7 along with standard deviations from probing experiments conducted in triplicate or greater for the following synthetic U2 snRNAs(A) RNA 1: WT, (B) RNA 3: C8A, (C) RNA 11: C21A. The arrows point to the location of mutated nucleotides (red) and their predicted base pairing partners (blue). Predicted secondary structure elements are highlighted by shading. BSL nucleotides are indicated by the pink bars, and reference stem loops by the blue bars. The right panels show secondary structure predictions by RNAstructure Fold based on chemical reactivity as represented by the color scale in the lower right. Mutations are highlighted by a purple circle. Percentages shown in green represent the probability of each structural element.

and while it explains the protections observed in the 5′ end of U2 snRNA, the low reactivity of nucleotides in the 3′ half of the BSL (nt 39–44) remains unaccounted for (Fig 4A). One issue with the structure prediction algorithm we used is the assumption of a single RNA conformation. We tried a second algorithm that considers the probability of multiple structures [48], but the top-ranking predictions all strongly resembled the previous prediction. Notably, the reactivity pattern for nucleotides 1–42 strongly resembles our observations for U2 snRNP in the absence of SF3A/SF3B complexes and protein-free U2 snRNA isolated from nuclear extracts.

When we examined the effects of single nucleotide mutations for base-paired nucleotides in Stem I, we often did not see the expected increase in reactivity in the altered base and its partner (Figs 4B and 4C and S2). Predicted structures based on the sequence and reactivity data often showed a shift in base pairing partners, but not a disruption of the stem. For example, mutating C8 in the lower half of Stem I to an A did not increase its reactivity, and results in a predicted partner swap of G25 for U27, along with the base-pairing shifts for neighbors U9

and G10 base pairs (Fig 4B). The most dramatic change in reactivity patterns occurred when C21 in the upper region of Stem I was changed to A, which results in the loss of the stem in the correlating predicted structures, which instead have slightly modified BSL structure that includes a four-nucleotide stem extension (Fig 4C). Combining the chemical mapping data and structure prediction models, we can conclude nearly all of the mutants maintain some form of an extended Stem I.

## Discussion

The unchanged chemical probing reactivities and protections within the different regions of the U2 snRNA show that the SF3B inhibitors have little impact on U2 snRNA structure in both the spliceosome and in the snRNP under the conditions that we tested. Both functional A-complex and SI-complex spliceosomes have reactivity patterns that indicate the presence of the upper region of Stem I, the branch helix and Stem IIa and IIb. This means that the BSL unwound, and that U2 snRNA BPRS engaged the intron in the branch helix before the SF3B inhibitor SSA impacted assembly. In the A-complex structure, the branch helix is held in place by SF3B1 in a closed conformation, in which the branch point adenosine is sequestered in a pocket formed between two SF3B1 HEAT repeats [8]. SF3B inhibitors occupy the same pocket and are proposed to block SF3B1 closing [41, 42]. The presence of the branch helix in the SI-complex suggests SF3B closing is not required for its formation. Instead, SF3B closing could signal that a proper branch helix is present to promote the next stage of spliceosome assembly. It will be interesting to chemically probe A-complex and SI-complex assembled on pre-mRNAs with branch point sequences that do not perfectly match the consensus sequence UACUAAC to see if strong base pairing alone stabilizes the branch helix in SI-complex, and to determine whether the same intron nucleotides are involved in base pairing interactions in both complexes. We also found that the top of Stem I, a region not observed in the cryo-EM structure of A-complex, remains intact in both complexes. This result means that branch helix formation and SF3B1 closure precedes Stem I unwinding. Together, our results indicate that SF3B inhibitors do not affect BSL unwinding or branch helix formation but that SF3B1 closure may be required for Stem I to be opened to allow engagement with U6 snRNA. Two caveats to this conclusion exist. First, chemical probing provides averaged information on the ensemble of our purified complexes, meaning that U2 snRNA structural fluctuations that occur on a faster time scale than our probing regime could still be affected by the SF3B inhibitors. Second, we have not tested whether the compounds influence the thermal stability of the complexes.

In the full U2 snRNP with SF3A and SF3B, the pattern of DMS modification of the U2 snRNA is consistent with the cryo-EM structure of the 17S U2 snRNP [6]. The absence of SF3A and SF3B complexes in the 12S U2 snRNP results in loss of Stem IIa, but surprisingly not the formation of Stem IIc. Because Stem IIc is present in the RNA alone, we conclude that either the remaining core U2 snRNP proteins or another factor in nuclear extract may interfere with Stem IIc formation. Such a factor could function in recycling of the U2 snRNP after spliceosome disassembly.

In both the 12S U2 snRNP and free RNA, protection from DMS at C5 near the 5′ end does fit canonical secondary structure models. Synthetic U2 snRNA constructs probed with 1m7 are also protected in the same region. When 1m7 reactivity is used as a restraint for RNA structure prediction, the RNAstructure algorithm favors an extended Stem I structure and leaves the somewhat less protected nucleotides in the 3′ half of the BSL as single stranded. Proximity and potential to form canonical base pairing interactions likely favor an extended Stem I over the BSL [46]. Alignment of U2 snRNA from a variety of species indicates that the potential to form an extended Stem I is conserved, although the requirement to base pair with U6 snRNA

in the spliceosome certainly contributes to sequence conservation. On the other hand, the BSL was also overlooked for many years because similar conservation was attributed to U6 snRNA base pairing needs. In the future, it will be important to probe U2 snRNA from sequences with divergent 5′ ends to see if the protections remain.

If U2 snRNA takes on an extended Stem I, the expectation is that nucleotides in the competing BSL structure will be unpaired and reactive to chemical probes. The relatively low reactivity of nucleotides in the BSL region for both synthetic and endogenous U2 snRNA indicates that the story may not be that simple. Understanding these dynamics is important because U2 snRNA's 5′ end because of the potential to impact both the biogenesis of and recycling of U2 snRNP when interactions with SF3A and SF3B are established. These rearrangements may also be coordinated with the switch from Stem IIc back to Stem IIa. Our results show that even with the beautiful snapshots provided by cryo-EM structures, there is still much to explore in the marvelous structural dynamics of U2 snRNA.

## Materials and methods

### HeLa nuclear extract

Nuclear extract was prepared from HeLa cells grown in DMEM/F-12 1:1 and 5% (v/v) newborn calf serum as previously described [49] with the exception that HEPES replaced Tris in all buffers. For high salt conditions, nuclear extract was dialyzed for four hours at 4˚C in a 10 kD cut-off Slide-A-Lyzer MINI Dialysis Device (Thermo Scientific) in buffer containing 20 mM HEPES pH 7.9, 420 mM potassium chloride, 0.2 mM EDTA, 20% glycerol, 0.5 mM dithiothreitol. Nuclear extract for A-complex assembly was depleted of U4 and U6 snRNAs using oligonucleotides complementary to the specified snRNAs and endogenous RNase H, as previously described in [36].

### Spliceosome complex purification

Spliceosome complexes were purified as previously described in Ilagan et al. 2015 with the following exceptions: A-complex and SF3B-inhibited spliceosome were assembled under *in vitro* splicing conditions in the appropriate HeLa nuclear extract with MS2:MBP tagged pre-mRNA substrate for 15 minutes and purified by size exclusion chromatography followed by amylose affinity selection. Purified splicing complexes were eluted in 150 mM potassium chloride, 5 mM EDTA, 20 mM HEPES pH 7.9, 1 mM dithiothreitol, 10 mM maltose.

### Chemical probing by DMS

U2 snRNP in nuclear extract prepared at 150 mM potassium chloride or 420 mM potassium chloride supplemented with additional HEPES (80 mM), magnesium acetate (2 mM) and was chemically modified with dimethyl sulfate (DMS, Sigma Aldrich) (0.7% v/v) for 5 minutes at room temperature. Reactions were quenched by adding BME to 37.4% and sodium acetate to 0.375 M. Purified spliceosomes in 90 mM potassium chloride, 3 mM EDTA, 33 mM HEPES pH 7.9, 8mM magnesium acetate, 0.6 mM dithiothreitol and 6 mM maltose were chemically modified with DMS (0.33%) at room temp for 10 minutes. Reactions were quenched by adding BME to 30% and 0.3 M sodium acetate.

### Chemical probing by 1m7

Purified spliceosomes in 90 mM potassium chloride, 3 mM EDTA, 33 mM HEPES pH 7.9, 8 mM magnesium acetate, 0.6 mM dithiothreitol, and 6 mM maltose were chemically modified

with 10 mM 1-Methyl-7-nitroisatoic anhydride (1m7, a gift from M. Ares) at room temperature for 10 minutes and then supplemented with.

## Primer extension by reverse transcription

Following chemical modification, RNA was purified by phenol:chloroform:iso-amyl alcohol (25:24:1) extraction, chloroform:iso-amyl alcohol (24:1) extraction and ethanol precipitation and resuspended in distilled H2O. Additionally, 10 picomoles of a DNA oligonucleotide complementary to nucleotides 97–119 of the U2 snRNA sequence was labeled with γ-32P ATP and purified by Sephadex G-25 (Sigma Aldrich) column. The RNA and labeled oligonucleotide were annealed by incubation at 95˚C for 2 min, 53˚C for 5 min, and on ice for 5 minutes and then added to reverse transcription reactions consisting of 50 mM Tris pH 8.0, 75 mM potassium chloride, 7 mM dithiothreitol, 1 mM dNTPs, 3 mM magnesium chloride and 80U of SuperScript III reverse transcriptase (Thermo Fisher Scientific). Reactions were incubated at 53˚C for 30 minutes, and then DNA was isolated by the addition of 0.3 M sodium acetate pH 5.2, 0.5 mM EDTA, and 0.05% SDS followed by ethanol precipitation. Sequencing ladders were generated from total RNA isolated from nuclear extract in similar reactions supplemented with a single ddNTP. DNA species were separated on a 9.6% (v/v) denaturing polyacrylamide gel that was dried onto Whatman paper and visualized by phosphorimaging.

## Chemical probing and analysis of synthetic U2 snRNAs

1m7 chemical probing of the synthetic RNA and data analysis was carried out essentially as described previously in [44, 50]. Briefly synthetic U2 snRNA templates were generated by PCR from overlapping oligonucleotides and used in T7 run-off transcription reactions. The transcribed RNA was separated on a 5% (v/v) denaturing polyacrylamide gel. Correlating bands identified by UV shadowing were excised, and the RNA isolated from excised bands by soaking gel slices in 0.3 M sodium acetate pH 4.8, 1 mM EDTA, 10% phenol overnight followed by ethanol precipitation.

For chemical probing, 0.6 pmol of RNA in 50mM Na-HEPES was heated to 95 ˚C for 3 minutes and allowed to fold at room temperature for 20 minutes followed by addition of 10 mM magnesium chloride. The RNA was aliquoted into a 96-well plate with 5 mM 1m7 or water and incubated for 10 minutes at room temperature followed by addition of oligo dT magnetic beads (Ambion) in 0.25 M Na-MES pH 6.0, 1.5 M NaCl, and 6.4 nM FAM-labeled primer. RNA/primers were isolated by magnetic bead immobilization, washed twice with 70% ethanol and resuspended in 2.5 μL water.

For reverse transcription, 20 units SuperScript III reverse transcriptase (Thermo Fisher Scientific), 5 mM dithiothreitol, 0.8 mM dNTPs, 50 mM Tris-HCl pH 8.3, 75 mM potassium chloride, and 3 mM magnesium chloride was added, and reactions incubated for 30 minutes at 48˚C. Ladders were generated by the addition of ddNTPs to additional control reactions. Following reverse transcription, 0.2 M sodium hydroxide was added, and samples were incubated for 3 minutes at 90˚C to degrade remaining RNA. Samples were neutralized by addition of 1.4 M sodium chloride and 0.6 M hydrogen chloride, followed by 1.3 M sodium acetate pH 5.2. DNA was purified by magnetic bead immobilization, washed twice with 70% ethanol, and resuspended in Hi-Di formamide supplemented with R0X350 dye standards (1:8 ROXF) (Thermo Fisher Scientific). 1:3 and 1:15 dilutions of the samples in ROXF were analyzed by capillary electrophoresis (ELIM Biopharmaceuticals).

Capillary electrophoresis data was analyzed with the HiTRACE MATLAB package [45]. Lanes of each RNA construct were aligned, and bands fit to Gaussian peaks that were background subtracted using the no-modification lane, corrected for signal attenuation, and

normalized to the internal hairpin control. The output is numerical array of reactivity values for each RNA nucleotide used as weights for structure prediction. Reactivity-guided secondary structure modeling was performed using the Biers MATLAB package (https://ribokit.github.io/Biers/) with the Fold function of the RNAstructure suite reactivity values applied as pseudoenergy modifiers to calculate the minimum free energy RNA structure. Bootstrapping analysis of reactivity-guided structure prediction was performed as described previously [44, 47]. Secondary structures were visualized using the VARNA applet [51].

## Supporting information

**S1 Fig. Chemical probing of U2 snRNA in U2 snRNP and RNA in the presence of SSA.** (A-B) Sequencing gel analysis of reverse transcription primer extension products from U2 snRNA isolated from nuclear extract with 17S U2 snRNP (150 mM KCl) or 12S snRNP (420 mM KCl) probed with DMS. Protein-free RNA purified from nuclear extract (RNA) was also probed. The gels are labeled like Fig 1, with lanes 9 showing a control reaction with DMSO and parallel sequencing reactions (lanes 1–2). Arrows point to primer extension products observed consistently across triplicate experiments. Nucleotides 1–46 are shown in A, and 47–95 in B. (C & D) Reactivity patterns mapped on two competing structural models of U2 snRNA with the structure most consistent with 17S U2 snRNP data shown on top. Nucleotides in grey correlate with primer extension stops in controls. The circles denote reactivity to DMS for 17S (green) or 12S (orange) U2 snRNPs or protein-free U2 snRNA (purple).
(PDF)

**S2 Fig. Chemical mapping of synthetic U2 snRNA mutants.** The left panels show normalized reactivities to 1m7 along with standard deviations from probing experiments conducted in triplicate or greater for the indicated synthetic U2 snRNA. The right panel shows secondary structure predictions by RNAstructure Fold based on chemical reactivity represented by the color relative to the lower right schematic. Mutations are highlighted in a purple circle. Percentages shown in green represent the probability of each structural element.
(PDF)

**S1 Raw images. Raw gel images and lane assignments for Figs 1 and 2.**
(PDF)

**S1 Table. 1m7 reactivities for synthetic U2 snRNA constructs.**
(XLSX)

## Acknowledgments

We gratefully acknowledge Manny Ares' generous gift of the 1m7 reagent, and we thank Christina Palka and Nicholas Forino for sharing protocols and expertise for the synthetic RNA probing experiments.

## Author Contributions

**Conceptualization:** Veronica K. Urabe, Arun K. Ghosh, Melissa S. Jurica.

**Formal analysis:** Veronica K. Urabe, Meredith Stevers, Melissa S. Jurica.

**Funding acquisition:** Arun K. Ghosh, Melissa S. Jurica.

**Investigation:** Veronica K. Urabe, Meredith Stevers.

**Methodology:** Veronica K. Urabe.

**Project administration:** Melissa S. Jurica.

**Resources:** Arun K. Ghosh.

**Supervision:** Melissa S. Jurica.

**Validation:** Veronica K. Urabe, Meredith Stevers, Melissa S. Jurica.

**Writing – original draft:** Veronica K. Urabe, Melissa S. Jurica.

**Writing – review & editing:** Veronica K. Urabe, Melissa S. Jurica.

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
