## [Decision Letter · Decision Letter 0]

13 Aug 2021

PONE-D-21-23018

U2 snRNA structure is influenced by SF3A and SF3B proteins but not by SF3B inhibitors

PLOS ONE

Dear Dr. Jurica,

Thank you for submitting your manuscript to PLOS ONE. After careful consideration, we feel that your manuscript is acceptable after minor revision. Therefore, we invite you to submit a revised version of the manuscript that addresses the points raised during the review process.

ACADEMIC EDITOR: Both reviewers thought your work acceptable. There are minor points that I am sure you should be able to address without any new experiments. Please submit your revised manuscript at your earliest. 

We look forward to receiving your revised manuscript.

Kind regards,

Ravindra N Singh, Ph.D.

Academic Editor

PLOS ONE

Journal Requirements:

5.In your Data Availability statement, you have not specified where the minimal data set underlying the results described in your manuscript can be found. PLOS defines a study's minimal data set as the underlying data used to reach the conclusions drawn in the manuscript and any additional data required to replicate the reported study findings in their entirety. All PLOS journals require that the minimal data set be made fully available. For more information about our data policy, please see http://journals.plos.org/plosone/s/data-availability.

Reviewers' comments:

Reviewer's Responses to Questions

**Comments to the Author**

1. Is the manuscript technically sound, and do the data support the conclusions?

Reviewer #1: Yes

Reviewer #2: Yes

2. Has the statistical analysis been performed appropriately and rigorously? 

Reviewer #1: N/A

Reviewer #2: N/A

3. Have the authors made all data underlying the findings in their manuscript fully available?

Reviewer #1: Yes

Reviewer #2: Yes

4. Is the manuscript presented in an intelligible fashion and written in standard English?

Reviewer #1: Yes

Reviewer #2: Yes

5. Review Comments to the Author

Reviewer #1: Urabe, Stevers, Ghosh, and Jurica determine and compare the chemical reactivity patterns of the U2 snRNA in a variety of physiological and medically-relevant contexts. They find that the SF3B1 inhibition with a representative small molecule made little difference in the U2 snRNA reactivity, in or out of the U2 snRNP or spliceosome assembly intermediate. Instead, an interesting finding is that the SF3A and/or SF3B proteins influence the structure of the 5´ end of the U2 snRNA. Indeed, they find a previously uncharacterized extension of Stem I. The work is solid and carefully interpreted. The results are important, especially considering that the dynamics of the U2 snRNA structure have been a long-sought and elusive topic in the field of pre-mRNA splicing. In addition, the results expand knowledge of the effects of SF3B1 inhibitors, which continue to be intriguing tools with potential anti-cancer applications.

Minor comments that could improve the manuscript:

1) The first sentence of the results extends 5.5 lines and would benefit from ‘divide and conquer’ revisions for clarity.

2) I’m curious whether an extension of BSL was predicted by either software, since 20-UAA-22 is complementary to 46-UUA-48 and I believe could explain the observed data as well as an extension of Stem I? Can this alternative be ruled out – what is the predicted difference in thermodynamic stability between the structure predictions? Note programs are available to predict the probability of pairing (e.g. Dave Mathews suite). Although I’m not certain chemical reactivity data can be included, results for the RNA alone could be informative.

3) The discussion would benefit from considering that the average equilibrium structure is detected here. Although the SF3B inhibitor has little impact on the average equilibrium chemical reactivity, the inhibitor could still influence dynamics beyond the limits of this experimental approach. Also, the thermal stability, which in turn would influence the energy requirement to dissociate the complex, has not been investigated here. These caveats should be expressed for the reader.

Reviewer #2: In “U2 snRNA structure is influenced by SF3A and SF3B proteins but not by SF3B inhibitors” Urabe et al probe the secondary structures of the U2 snRNA in early spliceosomal complexes (A-complex and SF3B-inhibited complex) using two different chemical probing methods. From their data, the authors predict the structure of the U2 snRNA bound to an intron as well as a part of the larger U2 snRNP. In general, their data shows that SF3B inhibitors do not alter the structure of U2 snRNA. They find that binding of U2 snRNA to protein components of the U2 snRNP does affect the structure as expected. In addition, they identify a previously uncharacterized extension of secondary structure at the 5’end of the U2 snRNA.

Overall, the paper is well written, and the data supports the conclusions drawn. While many of the results are not totally unexpected based on the spliceosome structures, the work is significant as it shows the dynamics of U2 structure and how it could (not) be affected by SF3B inhibitors. Following minor suggestions can further improve the clarity of the paper.

Minor comments:

• The figure legends for Figures 1 and 2 do not match the lane numbers in the figures. It seems like after lane 8 on both figures the lane numbers in the caption are increased by 1.

• In Figures 1 and 2 the grey bars and corresponding shading of the base-paired regions of the U2 snRNP would be easier to follow if they were different colors besides grey.

6. PLOS authors have the option to publish the peer review history of their article (what does this mean?). If published, this will include your full peer review and any attached files.

Reviewer #1: No

Reviewer #2: No

---

## [Author Response · Author response to Decision Letter 0]

26 Sep 2021

We are very grateful to the reviewers for their careful evaluation of our manuscript, and their suggestions for improvement. In addressing their concerns, we have rewritten portions of the introduction and discussion sections, and revised figures as requested.

Replies to specific concerns are given below.

Reviewer 1

Urabe, Stevers, Ghosh, and Jurica determine and compare the chemical reactivity patterns of the U2 snRNA in a variety of physiological and medically-relevant contexts. They find that the SF3B1 inhibition with a representative small molecule made little difference in the U2 snRNA reactivity, in or out of the U2 snRNP or spliceosome assembly intermediate. Instead, an interesting finding is that the SF3A and/or SF3B proteins influence the structure of the 5´ end of the U2 snRNA. Indeed, they find a previously uncharacterized extension of Stem I. The work is solid and carefully interpreted. The results are important, especially considering that the dynamics of the U2 snRNA structure have been a long-sought and elusive topic in the field of pre-mRNA splicing. In addition, the results expand knowledge of the effects of SF3B1 inhibitors, which continue to be intriguing tools with potential anti-cancer applications.

Minor comments that could improve the manuscript:

1) The first sentence of the results extends 5.5 lines and would benefit from ‘divide and conquer’ revisions for clarity.

We split the extended sentence into three separate statements to improve readability.

2) I’m curious whether an extension of BSL was predicted by either software, since 20-UAA-22 is complementary to 46-UUA-48 and I believe could explain the observed data as well as an extension of Stem I? Can this alternative be ruled out – what is the predicted difference in thermodynamic stability between the structure predictions? Note programs are available to predict the probability of pairing (e.g. Dave Mathews suite). Although I’m not certain chemical reactivity data can be included, results for the RNA alone could be informative. 

We were also surprised that the BSL was not predicted from our probing data, especially with mutants that destabilize the lower region of Stem I. Notably, A21 and A22 routinely showed high reactivity, indicating that they are likely not paired. 46-UUA-48 showed low reactivity and were always predicted to base pair with a complementary region in the “buffer” RNA preceding the last reference stem, which extends Stem IIC. We have subsequently changed those bases in the “buffer” RNA to prevent their pairing, but that alteration had no effect on the persistent prediction for the extended Stem I structure. Based on RNA sequence alone, the RNAStructure algorithm by the Mathews lab predicts the extended Stem I in the four top-ranking structures. It does, however, predict the presence of Stem IIa over Stem IIc in three of four structures. The BSL is never predicted, likely reflecting the need for a relatively unstable structure to allow for branch helix formation.

3) The discussion would benefit from considering that the average equilibrium structure is detected here. Although the SF3B inhibitor has little impact on the average equilibrium chemical reactivity, the inhibitor could still influence dynamics beyond the limits of this experimental approach. Also, the thermal stability, which in turn would influence the energy requirement to dissociate the complex, has not been investigated here. These caveats should be expressed for the reader.

We agree that these are important points to communicate, and we added the following sentences to the end of the first paragraph of the discussion:

“Two caveats to this conclusion exist. First, chemical probing provides averaged information on the ensemble of our purified complexes, meaning that U2 snRNA structural fluctuations that occur on a faster time scale than our probing regime could still be affected by the SF3B inhibitors. Second, we have not tested whether the compounds influence the thermal stability of the complexes.”

Reviewer 2

In “U2 snRNA structure is influenced by SF3A and SF3B proteins but not by SF3B inhibitors” Urabe et al probe the secondary structures of the U2 snRNA in early spliceosomal complexes (A-complex and SF3B-inhibited complex) using two different chemical probing methods. From their data, the authors predict the structure of the U2 snRNA bound to an intron as well as a part of the larger U2 snRNP. In general, their data shows that SF3B inhibitors do not alter the structure of U2 snRNA. They find that binding of U2 snRNA to protein components of the U2 snRNP does affect the structure as expected. In addition, they identify a previously uncharacterized extension of secondary structure at the 5’end of the U2 snRNA.

Overall, the paper is well written, and the data supports the conclusions drawn. While many of the results are not totally unexpected based on the spliceosome structures, the work is significant as it shows the dynamics of U2 structure and how it could (not) be affected by SF3B inhibitors. Following minor suggestions can further improve the clarity of the paper.

Minor comments:

• The figure legends for Figures 1 and 2 do not match the lane numbers in the figures. It seems like after lane 8 on both figures the lane numbers in the caption are increased by 1.

We have corrected the labeling of lanes for the figures in the figure legends.

• In Figures 1 and 2 the grey bars and corresponding shading of the base-paired regions of the U2 snRNP would be easier to follow if they were different colors besides grey.

We have added color to the bars as requested.

Editorial comments on journal requirements:

We made our best efforts to following the style and file naming requirements.

We are submitting the file as an MS word document. We will need support if a LaTeX file is required.

We updated the Funding Information to state:

Funding: This work was funded by National Institute of General Medical Sciences (NIGMS) of the National Institutes of Health (NIH) under award R01GM72649 to M.S.J and A.K.G. MS was supported by NIGMS T32GM13339 (UC Santa Cruz, Molecular, Cell and Developmental Biology Department Training Program) and V.K.U was supported by a National Science Foundation Graduate Research Fellowship under award DGE-1842400. The funders had no role in study design, data collection and analysis, decision to publish, or preparation of the manuscript. 

I am unable to find the ‘Financial Disclosure’ section.

4. PLOS ONE now requires that authors provide the original uncropped and unadjusted images underlying all blot or gel results reported in a submission’s figures or Supporting Information files. When you submit your revised manuscript, please ensure that your figures adhere fully to these guidelines and provide the original underlying images for all blot or gel data reported in your submission. In your cover letter, please note whether your blot/gel image data are in Supporting Information or posted at a public data repository, provide the repository URL if relevant, and provide specific details as to which raw blot/gel images, if any, are not available. 

We created a S3_raw_images.pdf containing full original gel images and a legend for all lanes. We also added a Supplemental Table with chemical reactivities for graphs in Fig 4 and S2 Fig. We deposited ABI files for high throughput chemical probing studies: 10.5281/zenodo.5527264

These changes are noted in an updated list of supplementary figures.

5.In your Data Availability statement, you have not specified where the minimal data set underlying the results described in your manuscript can be found. PLOS defines a study's minimal data set as the underlying data used to reach the conclusions drawn in the manuscript and any additional data required to replicate the reported study findings in their entirety. We will update your Data Availability statement to reflect the information you provide in your cover letter.

Data Availability: All relevant data are within the paper and its Supporting Information files. Raw data files for capillary electrophoresis analysis can be accessed from the following repository: 10.5281/zenodo.5527264

We have not cited any retracted articles.

---

## [Editor Report · Decision Letter 1]

30 Sep 2021

U2 snRNA structure is influenced by SF3A and SF3B proteins but not by SF3B inhibitors

PONE-D-21-23018R1

Dear Dr. Jurica,

We’re pleased to inform you that your manuscript has been judged scientifically suitable for publication and will be formally accepted for publication once it meets all outstanding technical requirements.

Kind regards,

Ravindra N Singh, Ph.D.

Academic Editor

PLOS ONE
---

## [Editor Report · Acceptance letter]

7 Oct 2021

PONE-D-21-23018R1 

U2 snRNA structure is influenced by SF3A and SF3B proteins but not by SF3B inhibitors 

Dear Dr. Jurica:

I'm pleased to inform you that your manuscript has been deemed suitable for publication in PLOS ONE. Congratulations! Your manuscript is now with our production department. 

Kind regards, 

on behalf of

Dr. Ravindra N Singh 

Academic Editor

PLOS ONE